# P-ADAPTERS: ROBUSTLY EXTRACTING FACTUAL INFORMATION FROM LANGUAGE MODELS WITH DIVERSE PROMPTS

**Benjamin Newman**[*]
Stanford University

**Prafulla Kumar Choubey**
Salesforce Research

**Nazneen Rajani**
Salesforce Research

## ABSTRACT

Recent work (e.g. LAMA (Petroni et al., 2019)) has found that the quality of the factual information extracted from Large Language Models (LLMs) depends on the prompts used to query them. This inconsistency is problematic because different users will query LLMs for the same information using different wording, but should receive the same, accurate responses regardless. In this work we aim to address this shortcoming by introducing *P-Adapters*: lightweight models that sit between the embedding layer and first attention layer of LLMs. They take LLM embeddings as input and output continuous prompts that are used to query the LLM. Additionally, we investigate Mixture of Experts (MoE) models that learn a set of continuous prompts ("experts") and select one to query the LLM. They require a separate classifier trained on human-annotated data to map natural language prompts to the continuous ones. P-Adapters perform comparably to the more complex MoE models in extracting factual information from BERT and RoBERTa while eliminating the need for additional annotations. P-Adapters show between 12-26% absolute improvement in precision and 36-50% absolute improvement in consistency over a baseline of only using natural language queries. Finally, we investigate what makes P-Adapters successful and conclude that a significant factor is access to the LLM's embeddings of the original natural language prompt, particularly the subject of the entity pair being queried.

## 1 INTRODUCTION

Recently, there has been an interest in meeting users' factual information needs using large language models (LLMs) as knowledge bases in place of traditional, index-based IR methods (Petroni et al., 2019; Metzler et al., 2021). In this conception of LLM knowledge bases, LLMs accumulate factual knowledge during pretraining, have their parameters frozen to maintain this knowledge, and then are queried by users using handcrafted, natural language prompts. For example, if one user wants to know what the capital of Canada is, they might query a frozen model with the prompt, "The capital of Canada is `[MASK]`.", while another might use different wording: "Canada, which has the capital city `[MASK]`." In order for LLMs to be effective knowledge bases, they have to be robust to different queries users could provide. Unfortunately, prior work has shown that LLMs are not robust: queries that are semantically equivalent may lead to inconsistent predictions (Jiang et al., 2020; Elazar et al., 2021). For example, taking the prompts above, the first prompt does extract the correct answer "Ottawa" from BERT Base, but the second extracts the incorrect "Winnipeg". This observation has led many works to try to find the optimal prompt or set of prompts for a given relation: the one(s) that will allow models to extract factual information the best (e.g., for the relation `capital_of`). These works have proposed ensembling sets of prompts (Jiang et al., 2020), optimizing which tokens are included in the prompt (Haviv et al., 2021; Shin et al., 2020), or forgoing the difficult discrete optimization problem entirely and instead optimizing the continuous embeddings that are input to the LLM (Li & Liang, 2021; Qin & Eisner, 2021; Zhong et al., 2021).

In our work, rather than focus on finding a single optimal prompt, we aim to help LLMs overcome this variability by *adapting* natural language prompts into continuous representations that allow

---

[*]`blnewman@cs.stanford.edu`. Work conducted during internship at Salesforce Research.

LLMs to accurately predict factual information. Because we are motivated by a user-focused setting, we want our adaptation method to require only a natural language prompt (e.g. "The capital of Canada is [MASK]") at inference time. No additional annotation for the relation (capital_of) between the subject ("Canada") and object ("Ottawa") of the entity pair in the prompt should be required. Nor should we need the identity of the subject separate from the prompt. At training time, our ideal method would only rely on (prompt, object) pairs (e.g. ("The capital of Canada is [MASK]", "Ottawa")), so collecting new data would not require additional annotation. Our research question is then as follows: in a setting where we are trying to prompt a frozen LLM using natural language alone, how can we encourage models to produce more consistent and accurate responses to potentially varied prompts.

We introduce a class of models called P-Adapters (short for "prompt adapters") to perform this adaptation, which satisfy our inference and training desiderata. P-Adapters sit between the embedding layer and the first attention layer of the frozen LLM, and modify the LLM's embeddings so factual information can be more effectively predicted (Figure 1). They are optimized end-to-end, only requiring (prompt, object) pairs at training time, and implicitly encourage consistency by learning to map variable training prompts to the same object.

We also investigate other models that could increase the consistency of natural language prompts: Mixture of Experts (MoE) models. Rather than adapt a natural language prompt, a MoE model maps it to an "expert" continuous prompt based on the relation between the prompt's subject and object. The MoE model then queries the frozen LLM with that expert. This method reduces variability by mapping all

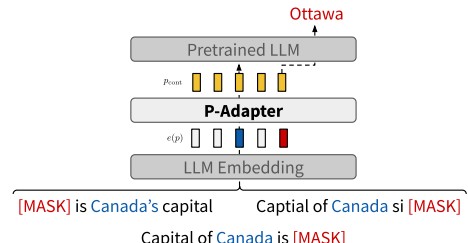

Figure 1: The P-Adapter Framework: P-Adapters sit between the embedding layer and the first attention layer of the LLM. They take as input the LLM embeddings and output continuous prompts that are fed into the LLM. The LLM is frozen (embeddings included), while the parameters of the P-Adapter are trained. The adapter helps mitigate variability among different phrasings and typographic errors in the prompts (as shown in the example inputs).

natural language prompts concerned with the same relation to a single prompt optimized for that relation. These methods require additional annotations during training and inference (namely the relation between the entities and the identity of the entity subject), and therefore do not conform to our desiderata. However, we include them in our discussion because they are useful for comparing to previous work and for understanding why P-Adapters are successful. We find that P-Adapter and MoE methods both strongly outperform a baseline of using the natural language prompts alone by between 12-28% on precision and 36-86% on consistency. While MoE models usually perform better, the small increase comes at the cost of requiring relation annotations at training time and specifying the subject of the entity at inference time. To simulate the variable kinds of natural language prompts we expect our methods to handle, we evaluate them in three out-of-distribution (OOD) settings: on a different set of natural language prompts, on a different distribution of entities (uniform rather than the distribution of Wikidata), and on natural language prompts with typographic errors. These OOD evaluations are especially important in light of recent work has found that models that appear to have knowledge of factual information often rely on surface-level feature associations. For example, models ignore negation, utilize stereotypical associations when making predictions (Kassner & Schütze, 2020; Poerner et al., 2020), and pick up on the distribution of objects from their training corpus rather than learn how to access arbitrary facts (Zhong et al., 2021; Cao et al., 2021). We find that the OOD distribution of entities is the most challenging setting, and while typographic errors reduce model performance, they do so by less than might be expected.

Finally, we investigate what makes P-Adapters effective, finding that it is important to keep some of the original natural language prompt embeddings available to the LLM. In particular, the availability of the subject of the prompt is the most important factor, in contrast with some other work suggesting the subject matters less (Cao et al., 2021).

Overall, we conclude that P-Adapters are helpful in reducing the impact of variable prompts, and are able to do so without requiring additional annotations. To encourage the use of P-Adapters to effectively extract factual information, we release the code used to train them. [1]

## 2 RELATED WORK

**Prompting Language Models for Factual Knowledge.** Prompting offers a low-parameter alternative to finetuning LLMs. In prompting, the LLMs are usually frozen and their pretraining task is used to fill in the desired information, which is usually a single token (Trinh & Le, 2018; Davison et al., 2019; Petroni et al., 2019). Cao et al. (2021) outlines three main approaches for prompting LLMs for factual information. The one we focus on here involves querying them zero-shot with a prompt that contains the single entity of interest (like in Figure 1) (Petroni et al., 2019).

Much previous work tries to lower-bound how much factual information LLMs store. Petroni et al. (2019) use one natural language prompt for each relation, while subsequent work argues that this approach underestimates LLM's abilities given how sensitive they are to input prompts. To support his argument, Jiang et al. (2020) and Elazar et al. (2021) generate paraphrases of prompts in the LAMA dataset and ensemble them to get a score for each relation. Others argue for optimizing discrete (Shin et al., 2020; Haviv et al., 2021) or continuous (Qin & Eisner, 2021; Zhong et al., 2021; Liu et al., 2021b) prompt for each relation instead of working with language ones. Elazar et al. (2021) thoroughly investigate prompt sensitivity in this setting, finding models often predict different entities for semantically equivalent prompts. Further, they propose continuing to train LLMs with a consistency loss function to improve their robustness. In this work, however, we focus on a user-inspired and lightweight approach that does not require updating LLM parameters.

**Robustness and Extracting Facts.** Training robust models—ones that are invariant to semantics-preserving input perturbations—is a challenge in many settings (Szegedy et al., 2013; Belinkov & Bisk, 2018) and has to led to theoretical and empirical innovations such as adversarial training (Sinha et al., 2017), provably robust models (Dvijotham et al., 2018), or more robust encoding schemes (Jones et al., 2020). Our works falls into this last category. In NLP, prior work investigates invariances to typos, word substitutions, irrelevant sentences, and syntactic perturbations (Jones et al., 2020; Alzantot et al., 2018; Jia & Liang, 2017; Iyyer et al., 2018).

For robustly extracting factual information, Poerner et al. (2020) and Kassner & Schütze (2020) argue that LLMs exploit surface form regularities when making predictions. Zhong et al. (2021) and Dufter et al. (2021) make a related observation, concluding that simpler models like randomly initialized LLMs, static embeddings, and even a Naive Bayes model can achieve a precision better than a majority baseline. Cao et al. (2021) argue that prompts that were found to do well in previous work overfit the distribution of objects in the training data rather than enabling knowledge extraction. They show that prompting with different sets of entities leads to very similar predictions. Therefore, we adopt an OOD evaluation setting to address these concerns. Low parameter finetuning methods (Houlsby et al., 2019; Ben Zaken et al., 2021; Liu et al., 2021a) have also been shown to have better OOD performance for generation and classification tasks (Li & Liang, 2021; Lester et al., 2021), which we hope to leverage for factual extraction here.

## 3 TERMINOLOGY

Taking inspiration from Liu et al. (2021a), we use the following terminology throughout this work. Each fact we want to extract consists of two parts: an *entity pair*—a *subject* ($x$) and an *object* ($y$)—and the *relation* ($r$) between them. We use the 41 relations in the LAMA dataset (Petroni et al., 2019). Each relation is expressed through a number of *templates* (e.g., one is "The capital of [X] is [Y]."). To query a language model for factual information, we substitute the subject in for the "[X]" in the template, and a [MASK] token into the object's place (the "[Y]") to form a *prompt* ($p$). We refer to this as a *natural language prompt* to emphasize that it consists of natural language tokens. Our prediction problem then consists of mapping $p$ to $y$.

---

[1] https://github.com/salesforce/factlm

LLMs perform this mapping by using their embedding layer, $e$, to embed $p$ into a sequence of continuous vectors, and then inputting this sequence to their first attention layer. We refer to this sequence of continuous vectors that are input to the LLM as a *continuous prompt* ($p_{\text{cont}}$). We make this distinction because P-Adapters sit between the embedding layer and the first attention layer of the LLM, so they take as input $e(p)$ and output a continuous prompt $p_{\text{cont}}$ that differs from $e(p)$. After passing $p_{\text{cont}}$ through the attention layers of the LLM, we take the LLM's prediction ($\hat{y}$) to be the token assigned the highest probability in the location of the [MASK] token in $p$, and compare it to $y$ to assess the model knowledge of the fact. Formally, a model is consistent if for any two prompts $p_1, p_2$ for the same $(x, y, r)$ triple, its top-1 predictions $\hat{y}_1$ and $\hat{y}_2$ are equal.

## 4 METHODS

To test extracting factual knowledge (rather than overfitting to training templates or entity pair distributions), we evaluate our P-Adapters in four settings:

1. **ID** templates and objects for testing in-domain generalization.
2. **OOD Prompts** for testing generalization to novel natural language prompts.
3. **OOD Objects** for testing whether our P-Adapter models learn to match the distribution of objects in the training set rather than predict arbitrary object entities.
4. **OOD Keyboard Errors** for testing robustness to typos in the natural language prompts.

### 4.1 DATA

**Entity Pairs.** We use the entity pairs and relations from the T-Rex split of the LAMA work (El-sahar et al., 2018; Petroni et al., 2019) in our experiments. This data includes 41 relations and approximately 1000 entity pairs for each relation sampled from Wikidata, including citations to where the relation manifests in Wikipedia. This data is used for evaluation. For training and validation, we use separate sets of entity pairs for each relation collected by Shin et al. (2020), which they use to optimize their discrete prompts. The entities pairs in the training, validation, and evaluation sets are all disjoint, though the distribution of object entities is similar. For the OOD Objects setting, we use the entity pairs in the uniform-wikidata dataset from Cao et al. (2021). This dataset was constructed so that each object appears the same number of times for each relation, in contrast to the ID evaluation set which contains a less uniform distribution of objects from Wikidata.

**Templates.** The templates we use are pooled from prior work: LAMA, LPAQA, and ParaRel datasets (Jiang et al., 2020; Elazar et al., 2021). LPAQA includes templates created automatically with a paraphrase model, mined from Wikipedia, and written by annotators, and ParaRel contains high-quality human-written templates. Additionally, we augment the ParaRel templates using the BERT lexical substitution system from Lee et al. (2021) by generating five paraphrases for each template, and then manually filtering out generations that do not preservens semantics. This gives us on average 81.4 prompt templates per relation (For more statistics, see Figure 6). We split the templates into two equal-sized groups: one for training and one for OOD Prompt evaluation. For the OOD Keyboard Errors setting, we use the training templates, except we introduce at least one typographic error in each template using the `nlpaug` package (Ma, 2019).

### 4.2 MODELS

First we introduce our P-Adapter models, and then we introduce our MoE models, and we finish with our oracle and baseline models. Recall that P-Adapters intercede between the LLM's embedding layer, $e$, and the first attention layer (Figure 1). We extract information from BERT Base Cased, BERT Large Cased, and RoBERTa Large (Devlin et al., 2018; Liu et al., 2019). Because these models' pretraining corpera contain Wikipedia, they should have access to the facts we test.

**P-Adapter.** P-Adapter models take as input the embeddings of the natural language prompt, $e(p)$, and output a new sequence of continuous embeddings, $p_{\text{cont}}$, that are used as input to the LLM in place of $e(p)$. Formally, a P-Adapter is a function $f_{\text{P-Adapter}} : e(p) \rightarrow p_{\text{cont}}$ trained to maximize

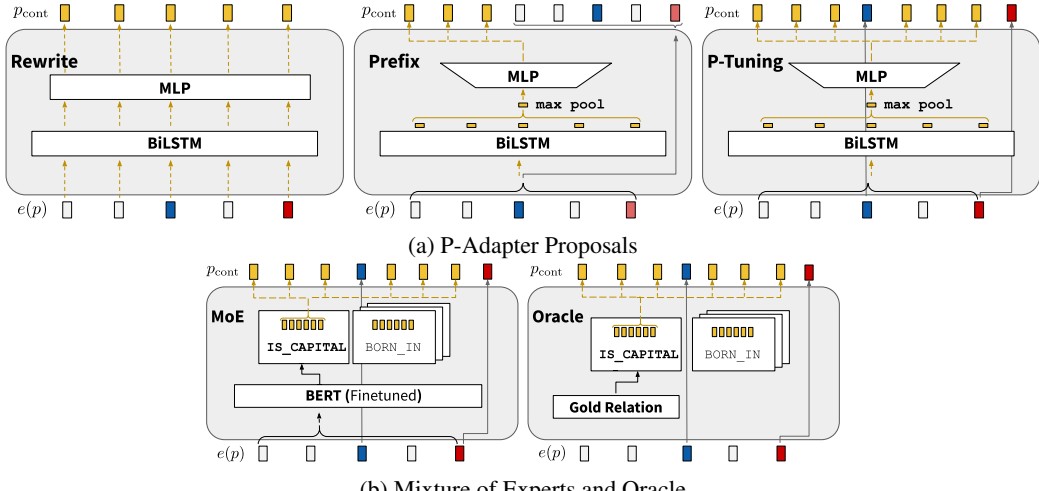

(a) P-Adapter Proposals

(b) Mixture of Experts and Oracle

Figure 2: P-Adapters lie in between the LLM Embeddings and the rest of the model (1). We propose end-to-end P-Adapters (2a) as well as a Mixture of Experts model (2b). Figures 2b and 2a are substituted into the "P-Adapter" block in 1. The subject embedding is in blue; the [MASK] embedding is red; embeddings generated by the P-Adapter are yellow; and other unmodified embeddings are gray. Dotted arrows represent inputs and outputs to model components and solid arrows represent copying from the input of the P-Adapter to its output.

$P_{\mathrm{LM}}(y \mid p_{\mathrm{cont}})$. The LLM's prediction is then:

$$\arg\max_{v \in \mathcal{V}} P_{\mathrm{LM}}\left(v \mid f_{\text{P-Adapter}}(e(p))\right),$$

where $\mathcal{V}$ is the LLM vocabulary. There are many different possible parameterizations of $f_{\text{P-Adapter}}$, and we describe three here. Two (Rewrite P-Adapter and Prefix P-Adapter) require no additional annotations, fitting the training and inference data criteria put forth in the introduction. The third (P-Tuning P-Adapter) does not fit our criteria, but we include it because it provides insight into what makes a P-Adapter effective (See Figure 2a for an illustration of each.)

The simplest P-Adapter we investigate is the **Rewrite P-Adapter**. The Rewrite P-Adapter is parameterized by a $d/2$-dimensional bidirectional LSTM followed by a single MLP applied to each hidden state independently (where $d$ is the LLM's hidden dimension). The output of the MLP, $p_{\mathrm{cont}}$, is input to the LLM. We use LSTMs to better compare our results with those of Liu et al. (2021b) who also use them, and because we want to use only a small number of parameters. Additionally, the prompts tend to be short and we do not do any pre-training, so LSTMs are effective.

Next we look at the **Prefix P-Adapter**. The Prefix P-Adapter is parameterized similarly to the Rewrite P-Adapter; however, its output is a prefix of embeddings that are prepended to $e(p)$. Like other works that learn prompt prefixes, the prefix has a fixed length (we use nine to compare with later models) (Li & Liang, 2021; Lester et al., 2021). Formally, using the Prefix P-Adapter, the input to the LLM is $[g(e(p)); e(p)]$, where $g$ is the application of a Bi-LSTM, a max pool, and an MLP to project the max pool's output into the size of the prefix. Unlike the Rewrite P-Adapter, the Prefix P-Adapter keeps all of the embeddings $e(p)$ accessible to the LLM's first attention layer.

Finally, we investigate the **P-Tuning P-Adapter**. This P-Adapter is based on work by Liu et al. (2021b) which presents a method called P-Tuning that learns a continuous prompt for each relation in the LAMA dataset. The method learns a function $g : r \to \mathbb{R}^{d \times 9}$. The $p_{\mathrm{cont}}$ consists of the output of $g$, the LLM's unmodified embedding of the subject, and the LLM's unmodified embedding of the mask token. It has the following form:

$$\left[g(r)_{[0:3]}; e(x); g(r)_{[3:6]}; e([\text{MASK}]); g(r)_{[6:9]}\right],$$

where bracket notation represents python-style list indexing.

The P-Tuning P-Adapter is implemented in a similar fashion, except $g$ takes as input $e(p)$ rather than $r$, and is parameterized similarly to the Prefix P-Adapter. Like the Prefix P-Adapter, the method out-

puts a fixed number of embeddings; however, unlike the Prefix P-Adapter, only the embeddings for the subject and `[MASK]` token are kept unchanged, rather than all of $e(p)$. Note that this P-Adapter additionally requires annotations for the identity of the subject during training and inference. This means that it does not fit the desiderata laid out in the introduction, but is included because it allows us to investigate what makes a P-Adapter successful.

**Mixture of Experts and Oracle.**  Enforcing consistency does not require modifying a natural language prompt like the P-Adapters do. One can also map each natural language prompt to a canonical continuous prompt, and use this prompt to query the frozen LLM. We explore this option next with our MoE and Oracle methods. In these, the canonical prompt is a continuous prompt optimized for the relation between the entities in the natural language prompt Liu et al. (2021b). These continuous prompts are learned using P-Tuning, and have the same format as described previously in the P-Tuning P-Adapters section; they ignore $e(p)$, depending only on the $x$, MASK, and $r$ (Liu et al., 2021b). We train these continuous prompts with P-Tuning using the same method and hyperparamters reported in Liu et al. (2021b), with one prompt for each of the 41 relations. In contrast to the P-Adapter methods, these require additional annotations with the relation of the prompt (See Figure 2b).

The **Mixture of Experts (MoE)** model consists of two components. The first is a classifier that predicts the relation between the entities of a natural language prompt. This classifier is a BERT Base Cased model finetuned on a 41-way classification task. The second component is a look-up table to map the predicted relations to the canonical continuous prompts.

Note that this is similar to a traditional mixture of experts model approach except that we do not use a weighted combination of prompts from different relations and instead just use the single prompt from the predicted relation.

The **Oracle** method is similar to the MoE approach, except rather than using a classifier to predict the relation, the gold relation is used at inference time. This method is an oracle because it achieves perfect consistency—it makes the same prediction for all prompts with the same entity pair. The oracle also serves as a direct comparison to the prior work of P-Tuning because P-Tuning assumes access to the gold relation.

**Baseline.**  Finally, the baseline is a model that takes as input the natural language prompt without any prefixes or optimization. Any effective and useful P-Adapter network would have to perform better than just using the natural language prompt itself.

### 4.3 METRICS

We report two metrics: precision@1 (P@1) and consistency. P@1 is a stringent measure of whether a prompt can extract a fact. It is the fraction of prompts where the correct object is the LLM's top prediction. Consistency measures whether a model's predictions for a given entity pair match across the different prompts $p$'s that contain the entities. We calculated consistency using the method from Elazar et al. (2021): the consistency of knowledge of a fact is the proportion of pairs of prompts for the fact where the model makes the same prediction. Formally, given a set of unordered pairs of prompts with the same entity pair, $P$, where there are $n$ unique prompts, and letting Top-1$(p) = \arg\max_{v \in \mathcal{V}} P_{\text{LM}}(v \mid f_{\text{prompt}}(e(p)))$ return the model's top prediction given a natural language prompt $x'$, the consistency is defined as:

$$\text{consistency}(\text{Top-1}, P) = \frac{\sum_{(p_i, p_j) \in P} \mathbb{1}[\text{Top-1}(p_i) = \text{Top-1}(p_j)]}{\frac{1}{2}n(n-1)}.$$

Note: a model can be consistent and inaccurate by always predicting the same incorrect object.

## 5  RESULTS

We present our main results for BERT Base in Table 1. The results for BERT Large and RoBERTa Large show similar trends and are available in the Appendix (Tables 2, 3, Figures 4 and 5; Qualitative Results are in Appendix D). Across all evaluation settings, we find optimized prompts lead to higher precision than natural language ones, as the low performance of the baseline indicates.

| Metric | P-Adapter | ID | OOD Prompts | OOD Objects | OOD KE |
|---|---|---|---|---|---|
| P@1 | Baseline | 0.157 | 0.157 | 0.069 | 0.092 |
| | Rewrite | $0.258 \pm 0.00$ | $0.247 \pm 0.00$ | $0.078 \pm 0.00$ | $0.167 \pm 0.00$ |
| | Prefix | $0.425 \pm 0.01$ | $0.415 \pm 0.01$ | $0.193 \pm 0.00$ | $0.326 \pm 0.01$ |
| | P-Tuning | $0.442 \pm 0.00$ | $0.422 \pm 0.00$ | $0.203 \pm 0.00$ | $0.325 \pm 0.00$ |
| | MoE | $0.488 \pm 0.01$ | $0.418 \pm 0.00$ | $0.237 \pm 0.02$ | $0.331 \pm 0.00$ |
| | Oracle | $0.496 \pm 0.01$ | $0.496 \pm 0.01$ | $0.238 \pm 0.02$ | $0.496 \pm 0.01$ |
| Consistency | Baseline | 0.126 | 0.133 | 0.097 | 0.068 |
| | Rewrite | $0.476 \pm 0.01$ | $0.448 \pm 0.02$ | $0.456 \pm 0.01$ | $0.223 \pm 0.01$ |
| | Prefix | $0.656 \pm 0.02$ | $0.613 \pm 0.02$ | $0.588 \pm 0.03$ | $0.452 \pm 0.03$ |
| | P-Tuning | $0.730 \pm 0.00$ | $0.646 \pm 0.01$ | $0.656 \pm 0.01$ | $0.476 \pm 0.01$ |
| | MoE | $0.947 \pm 0.03$ | $0.658 \pm 0.03$ | $0.916 \pm 0.05$ | $0.439 \pm 0.01$ |
| | Oracle | $1.000 \pm 0.00$ | $1.000 \pm 0.00$ | $1.000 \pm 0.00$ | $1.000 \pm 0.00$ |

Table 1: P@1 and Consistency for BERT Base across all our settings. P-Adapters are separated based on whether they require additional training/inference data. Note that the consistency of the Oracle is 1.0 and that the Baseline does not have any standard deviation because there is no optimization across runs. Results are microaveraged over all relations. (OOD KE: OOD Keyboard Errors).

**Precision.** Comparing the different evaluation settings, we observe the following. First, the OOD Objects was the most challenging setting, on average 20.41% lower precision than the ID setting, even for the oracle. However, the models performed similarly on the OOD Prompts as they did on the ID ones. At first this might seem to conflict with previous work that finds that the prompt has a large impact on performance (Jiang et al., 2020), but these results make claims about average rather than individual prompt performance. That said, precision is still higher ID than OOD, particularly for the MoE model. Looking at the MoE's relation classifier predictions, we can see it achieves an F1 of 99% for the ID and OOD Objects settings, but only 81% for OOD Prompts (See Appendix F). While this suggests some overfitting, our methods still outperform the baseline, so what they learn is still useful for the OOD natural language prompts. Finally, when evaluated on the OOD Keyboard Errors, we find that models still outperform the baseline. However, precision on the corrupted prompts is lower than the uncorrupted ones, and drops by a larger absolute percentage than the baseline does.

**Consistency.** Additionally, across all variation types we can see that optimized prompts lead to more consistency among the models' predictions. The consistencies dramatically increase from less than 0.2 for the baseline to over 0.4 (Table 1). Models were the least consistent on the OOD Keyboard Errors, and interestingly, were similarly consistent between the OOD Prompts and OOD Objects evaluations despite having a much higher precision for the OOD Prompts. The oracle has a perfect consistency of 1.0, because it uses the same continuous prompt for all facts with the same relation, so the predictions for that fact are all the same. The MoE model has a high consistency for a similar reason: as long as the classifier predicts two prompts for the same entity pair come from the same relation, the predictions will be the same, and therefore consistent. The P-Tuning P-Adapter often has a a high consistency as well, sometimes even higher than the MoE. This suggests that in P-Tuning P-Adapter, the main factor driving the model's prediction is the subject of the entity pair, because the unmodified embedding of the subject is shared among all of the prompts for the same object.

In the P-Adapters, this consistency is implicitly encouraged by the current training procedure by training P-Adapters to map from various natural language prompts to the same object. We investigate adding an additional loss term to explicitly encourage consistency and find that doing so increases consistency by 5% on average. This suggests that implicitly encouraging consistency can be helpful but is not necessary. The procedure and results are detailed in Appendix E.

**Unmodified Embeddings.** We observe that giving the LLM access to its unmodified embeddings of the natural language prompt was helpful. The Prefix P-Adapter and the P-Tuning P-Adapter, which have access to some or all of these embeddings, achieve higher precisions than the Rewrite P-Adapter, which alters all of them. Because of this, we perform additional experiments to see what part of natural language prompt was important to keep.

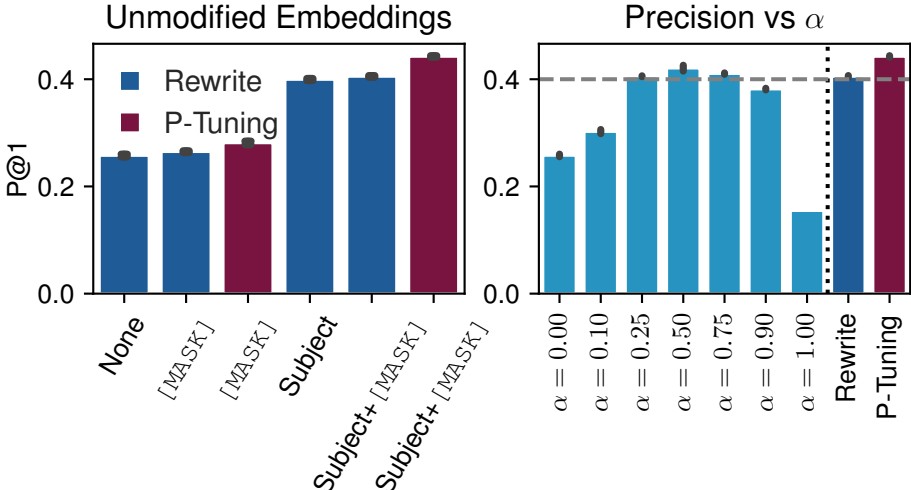

Figure 3: The above results show which of the LLM's embeddings are important to keep unmodified. On the left, the x-axis has the embeddings that are not modified by the P-Adapter (e.g. "None" is the Rewrite P-Adapter, where all embeddings are modified). These results are computed with BERT Base and the ID data. In the plot on the right, we compare different $\alpha$'s to the P-Tuning P-Adapter and the version of the "Rewrite" P-Adapter with the unmodified subject and [MASK] embeddings, finding the $\alpha = 0.5$ performs as well as the other adapters that require additional annotation.

One hypothesis is that the [MASK] token embedding should remain unmodified, and the Rewrite P-Adapter performs poorly because it corrupts this embedding. We investigate this in two ways. First, we train a version of the Rewrite P-Adapter that replaces the P-Adapter's output in the [MASK] token position with the LLM's [MASK] token embedding. Second, we train a version of the P-Tuning P-Adapter whose output only has the LLM's [MASK] embedding, not the subject embedding. However, we find these do not increase performance much compared to the Rewrite P-Adapter.

Another hypothesis is that the subject embeddings must remain unmodified, and that the Rewrite P-Adapter corrupts the subject, leading to lower precision. To address this, we train a version of the Rewrite P-Adapter model that replaces the embeddings for the subject in the P-Adapter output with the LLM's embeddings for the subject. We find this is much more effective, increasing precision by $13\%$. For comparison with the P-Tuning P-Adapter, which keeps the subject and [MASK] embeddings the same, we also train a version of the Rewrite P-Adapter that replaces the embeddings at both the subject and [MASK] token positions with the LLM's embeddings and find that this performs only $4\%$ worse than the P-Tuning P-Adapter. The results are visible in Figure 3. From these experiments, we conclude that the most important aspect of the prompt is the subject token. This also helps explain why the Prefix P-Adapter does well: it has access to the LLM's embeddings for the entire natural language prompt, which includes the subject.

Unfortunately, while these changes to the Rewrite P-Adapter models show promising results, they require knowing the index of the subject tokens at training and inference time, which contradicts our requirement to not use extra annotations. Fortunately, there is another way to incorporate the unmodified LLM's embeddings of the natural language prompt into the output of the P-Adapter: we can interpolate between the P-Adapter output and the unmodified embeddings. To do this, we train a third type of Rewrite P-Adapter. If $f_{\text{rewrite-adapter}}$ is the original Rewrite P-Adapter, the equation for the new Rewrite P-Adapter is:

$$f_{\text{P-Adapter}}(p) = \alpha e(p) + (1 - \alpha) f_{\text{rewrite-adapter}}(e(p)).$$

where $\alpha$ is a hyperparameter. When $\alpha = 0$, this P-Adapter is equivalent to the original Rewrite P-Adapter, and when $\alpha = 1$ it is the baseline. We test $\alpha \in \{0.1, 0.25, 0.5, 0.75, 0.9\}$, and find $\alpha = 0.5$ to perform the best. It outperforms the Rewrite P-Adapter when the subject and [MASK] tokens are substituted in, though barely underperforms compared to the P-Tuning P-Adapter (See Figure 3).

## 6 DISCUSSION

Our goal was to create models to adapt arbitrary user queries so predictions from frozen LLMs were accurate and consistent. One desiderata was to train with only (prompt, object) pairs, so that a user's experience would match current IR system experiences. Despite this requirement, some of the methods we introduce do use additional information—for comparison to other methods and understanding our own. For example, the P-Tuning P-Adapter and the MoE models both need the identity of the subject of the user's prompt, and requiring a user to provide this feels cumbersome and redundant. One could obviate this need using an off-the shelf NER system as well, and we leave exploring this to future work.

Additionally, the MoE models train their classifiers using annotations for the relation between the entities in the prompts. We use these annotations to compare to previous work: previous work assumes that relation is the most useful intermediate variable for factual extraction, but this does not have to be the case. For example, good prompts across different relations like "Ottawa is the capital of [MASK]" and "The second largest country by land area is [MASK]" might share components even though they do not share a relation. Our P-Adapters allow for greater flexibility by conditioning on the natural language prompt itself rather than the relation. Importantly, while using extra information leads to better results in some cases, the Prefix P-Adapter usually achieves comparable precision. And while the Rewrite P-Adapter flounders, ensembling its outputs with the LLM embeddings leads to greater success.

We speculate that there are two likely reasons for the P-Adapters' success. First, not updating the LLM parameters likely maintains factual knowledge learned during pretraining—Elazar et al. (2021) show that continuing to finetune the models on extracting factual information can lead to lower accuracy. Second, the natural language prompts are likely close to being effective prompts, so small changes from P-Adapters are sufficient to elicit the correct objects.

It is also worth noting that while the Prefix P-Adapter approaches the performance of the techniques with more information, this performance is still quite modest. The oracle was able to achieve only a precision of $50\%$ in the cases with ID objects, missing half of the facts. For the OOD Objects evaluation set, it performed much worse, only correctly predicting about $25\%$ of the facts. While the P@1 metric is quite strict, these numbers do not bode well for this approach. Cao et al. (2021) blame this poor performance on the optimized prompts overfitting to the training object distribution, and they observe that when the subjects are changed, the LLM's distributions do not change all that much. They even observe that the LLM's distributions when prompts have subjects are similar to those when the subjects are completely omitted. Our findings point to a more nuanced result. While it might be true that the LLM's distributions in these cases are correlated, the subject does matter when considering the top prediction. We find that without the LLM's embeddings of the subject accessible, the P-Adapters performs even more poorly.

## 7 CONCLUSION

LLMs implicitly learn factual information during pretraining, but accurately and consistently extracting this information is challenging. Our work complements previous work on extracting factual information from LLMs by introducing a more user-oriented setting: we want our models to only take in varied natural language prompts, and return the objects that meet information needs. To do this, we propose P-Adapter models, which sit between the embedding layer and the first attention layer of LLMs. These are effective compared to other methods that require additional annotations (e.g., the relation between the entities in the prompt), and we perform ablations determining that their success is due to keeping the LLM's embeddings of the subject available to the LLM. While we focus on the task of extracting factual information from LLMs, P-Adapters potentially provide a general framework for adapting to variable inputs, for example in reading comprehension or dialogue. While there is still room for the further improvement in using LLMs as knowledge bases, and we see P-Adapters as an important part of the future of this endeavor.

## 8 ETHICS STATEMENT

While our work is inspired by how users interact with current IR systems, we do not perform any experiments with human subjects. Additionally, our method improves the ability to extract factual information from pretraining corpora, so this means it may be possible to exploit our method to elicit the biased or harmful predictions from these corpora as well, though we do not study this. Finally, improving the ability to accurately and consistently extract information from text sources has privacy concerns if LLMs are pretrained on private or otherwise sensitive data. For the LLMs we study, this is not a concern as they were trained on Wikipedia and Books Corpus data rather than web data.

## 9 REPRODUCIBILITY STATEMENT

The descriptions of our models and training procedure in Section 4 and the code available at the link below is sufficient for reproducing our results. The data we use is available in the Github repositories of the work we cite that creates these resources. For running experiments, see the README.md file here: `https://github.com/salesforce/FactLM`.

ACKNOWLEDGMENTS

The authors would like to thank Wenhao Liu, John Hewitt, and Alex Tamkin for their valuable discussions and feedback. We would also like to thank our reviewers for suggesting additional experiments and analyses to strengthen the claims made in the work.

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

# A  ALL RESULTS

See Figures 4 and 5 as well as Tables 2 and 3.

| LLM | P-Adapter | ID | OOD Prompts | OOD Objects | OOD KE |
|---|---|---|---|---|---|
| BERT Base | Baseline | 0.157 | 0.157 | 0.069 | 0.092 |
| | Rewrite | $0.258 \pm 0.00$ | $0.247 \pm 0.00$ | $0.078 \pm 0.00$ | $0.167 \pm 0.00$ |
| | Prefix | $0.425 \pm 0.01$ | $0.415 \pm 0.01$ | $0.193 \pm 0.00$ | $0.326 \pm 0.01$ |
| | P-Tuning | $0.442 \pm 0.00$ | $0.422 \pm 0.00$ | $0.203 \pm 0.00$ | $0.325 \pm 0.00$ |
| | MoE | $0.488 \pm 0.01$ | $0.418 \pm 0.00$ | $0.237 \pm 0.02$ | $0.331 \pm 0.00$ |
| | Oracle | $0.496 \pm 0.01$ | $0.496 \pm 0.01$ | $0.238 \pm 0.02$ | $0.496 \pm 0.01$ |
| BERT Large | Baseline | 0.163 | 0.160 | 0.077 | 0.099 |
| | Rewrite | $0.117 \pm 0.06$ | $0.115 \pm 0.06$ | $0.042 \pm 0.02$ | $0.072 \pm 0.03$ |
| | Prefix | $0.431 \pm 0.03$ | $0.424 \pm 0.02$ | $0.207 \pm 0.01$ | $0.337 \pm 0.02$ |
| | P-Tuning | $0.470 \pm 0.00$ | $0.443 \pm 0.00$ | $0.222 \pm 0.00$ | $0.346 \pm 0.01$ |
| | MoE | $0.498 \pm 0.01$ | $0.431 \pm 0.01$ | $0.250 \pm 0.02$ | $0.339 \pm 0.01$ |
| | Oracle | $0.510 \pm 0.02$ | $0.509 \pm 0.02$ | $0.258 \pm 0.03$ | $0.510 \pm 0.02$ |
| RoBERTa Large | Baseline | 0.122 | 0.121 | 0.054 | 0.071 |
| | Rewrite | $0.222 \pm 0.04$ | $0.214 \pm 0.04$ | $0.068 \pm 0.01$ | $0.148 \pm 0.03$ |
| | Prefix | $0.385 \pm 0.03$ | $0.385 \pm 0.03$ | $0.179 \pm 0.02$ | $0.311 \pm 0.02$ |
| | P-Tuning | $0.360 \pm 0.07$ | $0.359 \pm 0.06$ | $0.167 \pm 0.03$ | $0.306 \pm 0.03$ |
| | MoE | $0.335 \pm 0.11$ | $0.290 \pm 0.09$ | $0.156 \pm 0.05$ | $0.231 \pm 0.06$ |
| | Oracle | $0.340 \pm 0.11$ | $0.340 \pm 0.11$ | $0.157 \pm 0.06$ | $0.340 \pm 0.11$ |

Table 2: Tabular format showing precision from Figure 4. The Baseline does not have any standard deviation because there is no randomness across runs. Results are microaveraged over all relations. (OOD KE: OOD Keyboard Errors)

# B  TEMPLATES

See Figure 6 and Table 4.

| LLM | P-Adapter | ID | OOD Prompts | OOD Objects | OOD KE |
|---|---|---|---|---|---|
| BERT Base | Baseline | 0.126 | 0.133 | 0.097 | 0.068 |
| | Rewrite | $0.476 \pm 0.01$ | $0.448 \pm 0.02$ | $0.456 \pm 0.01$ | $0.223 \pm 0.01$ |
| | Prefix | $0.656 \pm 0.02$ | $0.613 \pm 0.02$ | $0.588 \pm 0.03$ | $0.452 \pm 0.03$ |
| | P-Tuning | $0.730 \pm 0.00$ | $0.646 \pm 0.01$ | $0.656 \pm 0.01$ | $0.476 \pm 0.01$ |
| | MoE | $0.947 \pm 0.03$ | $0.658 \pm 0.03$ | $0.916 \pm 0.05$ | $0.439 \pm 0.01$ |
| | Oracle | $1.000 \pm 0.00$ | $1.000 \pm 0.00$ | $1.000 \pm 0.00$ | $1.000 \pm 0.00$ |
| BERT Large | Baseline | 0.122 | 0.123 | 0.094 | 0.069 |
| | Rewrite | $0.550 \pm 0.22$ | $0.538 \pm 0.23$ | $0.571 \pm 0.22$ | $0.399 \pm 0.29$ |
| | Prefix | $0.619 \pm 0.07$ | $0.583 \pm 0.04$ | $0.566 \pm 0.06$ | $0.446 \pm 0.02$ |
| | P-Tuning | $0.759 \pm 0.01$ | $0.656 \pm 0.01$ | $0.694 \pm 0.02$ | $0.484 \pm 0.02$ |
| | MoE | $0.955 \pm 0.03$ | $0.680 \pm 0.04$ | $0.933 \pm 0.04$ | $0.456 \pm 0.03$ |
| | Oracle | $1.000 \pm 0.00$ | $1.000 \pm 0.00$ | $1.000 \pm 0.00$ | $1.000 \pm 0.00$ |
| RoBERTa Large | Baseline | 0.093 | 0.096 | 0.076 | 0.050 |
| | Rewrite | $0.438 \pm 0.03$ | $0.403 \pm 0.02$ | $0.421 \pm 0.02$ | $0.217 \pm 0.01$ |
| | Prefix | $0.540 \pm 0.07$ | $0.531 \pm 0.06$ | $0.491 \pm 0.08$ | $0.425 \pm 0.04$ |
| | P-Tuning | $0.731 \pm 0.13$ | $0.707 \pm 0.15$ | $0.702 \pm 0.18$ | $0.600 \pm 0.21$ |
| | MoE | $0.974 \pm 0.01$ | $0.661 \pm 0.03$ | $0.970 \pm 0.02$ | $0.436 \pm 0.03$ |
| | Oracle | $1.000 \pm 0.00$ | $1.000 \pm 0.00$ | $1.000 \pm 0.00$ | $1.000 \pm 0.00$ |

Table 3: Tabular format for the consistency from Figure 5. The Baseline does not have any standard deviation because there is no randomness across runs. Note that the Oracle models achieves a consistency of 1. Results are microaveraged over all relations. (OOD KE: OOD Keyboard Errors)

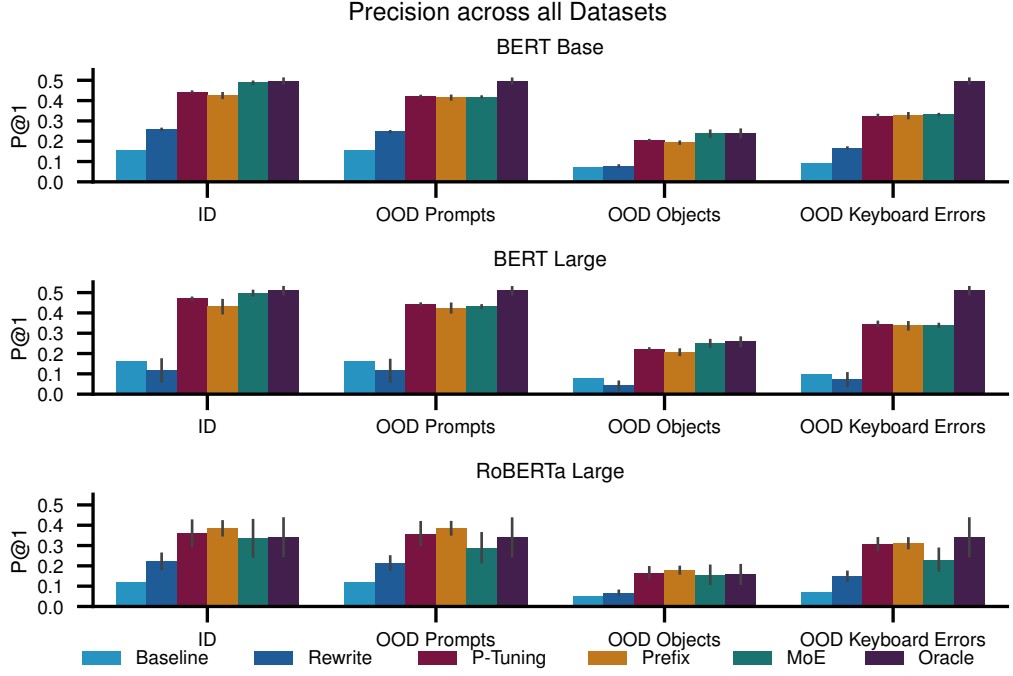

Figure 4: Results across our ID and OOD datasets across three LLMs. Each row is a different model, the x-axis is the dataset and error bars show standard deviation across either 3 runs. We can see that the oracle method outperformed all others, while the baseline and rewriter did the worst. The trends are similar across all models—the only difference come from some RoBERTa MoE models having higher variance than others which is explained by some runs not finding the best P-Tuning templates.

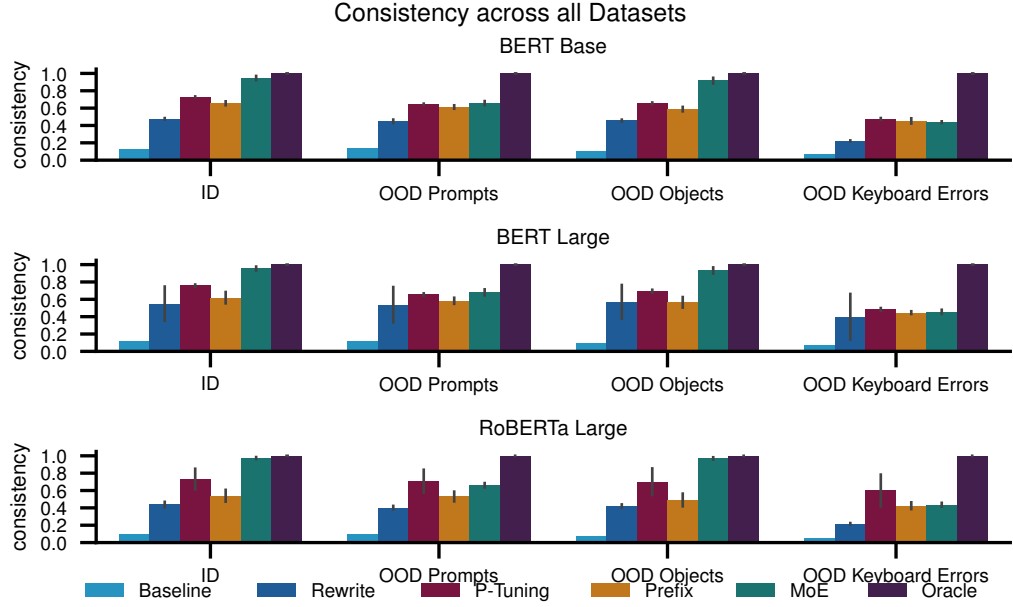

Figure 5: Consistency across all LLMs and evaluation sets.

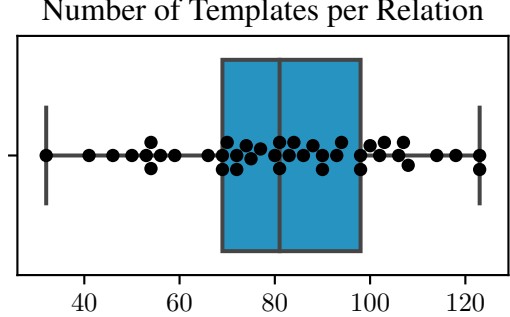

Figure 6: Distribution of number of templates across the 41 relations we tested. Minimum is 32, maximum is 123 and mean is 81.44.

| | |
|---|---|
| subject | Canada |
| object | Ottawa |
| relation | IS_CAPITAL |
| template | The capital of [X] is [Y]. |
| natural language prompt | The capital of Canada is [MASK]. |
| prediction | Toronto |

Table 4: Here the terminology is applied to the example from the introduction. Note that the model is incorrect in this example.

## C  TRAINING DETAILS

All of our P-Adapters were trained using the hyperparameters from Liu et al. (2021b): Adam optimizer with a learning rate of $1e-5$, weight decay of $5e-4$, a batch size of $128$, and an exponential learning rate decay schedule with a decay rate of $0.98$ (Kingma & Ba, 2015). Each result is the mean over five random seeds.

Our MoE classifiers were trained using an AdamW optimizer with a learning rate of $0.001$ and linear learning rate decay (Loshchilov & Hutter, 2018). We use HuggingFace Transformers to train the model for 3 epochs on the same training data used to train the P-Adapter models (Wolf et al., 2020). Each result is the mean over three random seeds rather than five because finetuning the classifier was more costly than training the the P-Adapters.

## D  QUALITATIVE ANALYSIS

We also performed qualitative analysis to determine how P-Adapters' predictions compare to the Baselines' and MoE models'. We performed this analysis using the BERT base models, and focused on the Prefix P-Adapter.

In general, we find that the Baseline tends to include more nonsensical or untopical vocabulary items. For example, for P36, the `capital_of` relation, the baseline predicts the capital of "Iraq" to be '|', 'She', and 'It' for some prompts. There are also situations where despite this variability, the baseline fails to predict the correct object for any of the prompts. For example, for the relation `official_language_of`, there are 60 natural language prompts. The correct object of the subject "Raahe" is "Finnish", and it does not appear in the baseline's 28 distinct predictions. However, half of the P-Adapter's predictions are the correct "Finnish".

While the P-Adapter's predictions are more consistent than the baseline's, they are less consistent than the MoE's. While this is undesirable from the perspective of consistency, it can lead to greater accuracy in some cases. For example, for the relation `language_spoken`, the MoE model predicted the object "French" for the subject "Guy Maddin" for 100% of the prompts, while the variability in P-Adapters allowed them to predict the only correct answer, "English" in $89.3\%$ of the prompts (and "French" in $9.3\%$ of them).

There are downsides to this flexibility, however. We also observe examples where P-Adapters are tricked by the surface form of the prompt. For example, for the relation `place_of_death`, "Akihiko Saito" is predicted be Tokyo by $95\%$ of phrases, even though the correct answer is "Iraq". There are also examples where P-Adapters' (particularly, the Prefix P-Adapters') predictions appear to be copied from the input where the MoE models' are not. For example, for prompts like "Windows Server 2003 was manufactured by `[MASK]`." P-Adapters correctly predict "Microsoft" for $87\%$ of paraphrases, but they also predict "Windows" $13\%$ of the time, which we do not observe the MoE models doing.

## E  EXPLICITLY OPTIMIZING FOR CONSISTENCY

Currently, P-Adapters encourage consistency in LLMs' predictions implicitly, as the loss encourages LLMs to predict the same, correct object for variable natural language inputs. However, one could imagine that explicitly encouraging consistency could be helpful. One way to explicitly train for consistency is by encouraging not just the top-1 prediction to be the same, but the distribution over objects to the be the same for different natural language prompts for the same fact (Elazar et al., 2021). To do this, we train with pairs of natural language prompts for the same fact, and encourage the distributions over the predicted objects to be similar using the symmetric KL-Divergence. Formally, given two prompts for the same entity pair and relation $p_1, p_2$, and an LM $P_{LM}$ we define our loss as:

$$\ell_{\text{consistency}} = \mathcal{D}_{KL}(P_{LM}(p_1) \,||\, P_{LM}(p_2)) + \mathcal{D}_{KL}(P_{LM}(p_2) \,||\, P_{LM}(p_1))$$

We then add this to the loss, $\ell$ that we were previously calculating—the cross-entropy loss between the distribution over the object and the correct object, giving a total loss of:

$$\ell_{total} = \ell + \lambda \, \ell_{\text{consistency}}$$

where we use $\lambda = 0.5$.

This approach is inspired by the approach taken by Elazar et al. (2021), with the differences being that they finetune all of the parameters of the LLM, calculate this loss over all prompts with the same fact, and also find that in their setting they need to additionally continue with masked-language modeling training.

We train new Prefix-P-Adapters on the BERT Base with this new loss, and find that the consistency increases slightly, and the P@1 stays the same or slightly decreases. This boost in consistency is beneficial, but the small increase (compared to the baseline) does not suggest that explicitly encouraging consistency is necessary (See Table 5).

|  | ID | OOD Prompts | OOD Objects | OOD KE |
|---|---|---|---|---|
| *P@1* |  |  |  |  |
| Baseline | 0.157 | 0.157 | 0.069 | 0.092 |
| Prefix P-Adapter | $0.425 \pm 0.011$ | $0.415 \pm 0.009$ | $0.193 \pm 0.004$ | $0.326 \pm 0.012$ |
| + consistency | $0.415 \pm 0.002$ | $0.408 \pm 0.001$ | $0.190 \pm 0.002$ | $0.326 \pm 0.001$ |
| MoE | $0.488 \pm 0.007$ | $0.418 \pm 0.002$ | $0.237 \pm 0.020$ | $0.331 \pm 0.003$ |
| *Consistency* |  |  |  |  |
| Baseline | 0.126 | 0.133 | 0.097 | 0.068 |
| Prefix P-Adapter | $0.656 \pm 0.023$ | $0.613 \pm 0.020$ | $0.588 \pm 0.026$ | $0.452 \pm 0.031$ |
| + consistency | $0.680 \pm 0.008$ | $0.645 \pm 0.007$ | $0.628 \pm 0.005$ | $0.507 \pm 0.010$ |
| MoE | $0.947 \pm 0.034$ | $0.658 \pm 0.032$ | $0.916 \pm 0.046$ | $0.439 \pm 0.013$ |

Table 5: P@1 and Consistency of Prefix P-Adapters on BERT Base. +consistency indicates training with the consistency loss (averaged over three runs). Consistency increases a small amount, while accuracy stays the same. The Baseline and MoE results are included for reference.

## F  RELATION CLASSIFIER

We finetune BERT-base-cased to predict the relation from the natural language prompt for the MoE models. The F1 scores are microaveraged across three random seeds are visible in Table 6.

| ID | OOD Prompts | OOD Objects | OOD KE |
|---|---|---|---|
| $0.989 \pm 0.004$ | $0.812 \pm 0.011$ | $0.990 \pm 0.003$ | $0.607 \pm 0.015$ |

Table 6: Microaveraged F1 score for the relation classifier component of the MOE models in each of the evaluation settings we investigate.

