# OpenReview forum: "P-Adapters: Robustly Extracting Factual Information from Language Models with Diverse Prompts"
_ICLR.cc/2022/Conference — ICLR 2022 Poster_

### Official Review · Reviewer_anXM · 2021-10-29

**Correctness:** 3
**Technical Novelty And Significance:** 3
**Empirical Novelty And Significance:** 2
**Recommendation:** 6
**Confidence:** 4

**Main Review:**

+ves:
+ Overall the paper is clear
+ Use of LLM for extracting factual information is well motivated
+ The motivation behind the use of natural language prompts to improve the user experience is generally clear. However, it can benefit from further citation of the literature in that domain. Having a few citations of works that compare the user experience with and without the additional information will make your argument in the paper stronger.
+ Empirical results  suggest that P-Adapters improve the consistency


-ves:
- research questions are dispersed all over the paper. What I mean by this:

    Abstract: you say that you tackle the inconsistency problem and you want to use a lightweight approach. Hence, I can see that there are two research questions: 1) inconsistency of LLM and 2) efficiency - how to tackle the inconsistency problem with a light weight model.

    Introduction: you say that focus is the continuous prompts and user-friendly prompts. These are another two research questions.

    Discussion: you hypothesising that the success of P-Adapter is due to not updating the LLM parameters. Such that it is likely to maintain all of factual knowledge learned during pretraining. Which to me is yet another research question - addressing the inconsistency of LLM with frozen weights vs fine-tuning. In Introduction you talk about the frozen weights of LLM but do not motivate why you decide to keep them frozen until the Discussion section.

   Can you please state and motivate clearly your (full) research question somewhere in Introduction?

- Lack of experiments:
    1) (Minor) Why didn’t you train BERT fully with the proposed natural language prompts. You emphasise the importance of this experiment yourself in the Discussion section.

    2) (Minor) Why didn’t you train (fine-tune) BERT with P-Adapters? This experiment will also allow you to answer your hypothesis in the Discussion section.

    3) (Major) Did you try to freeze the parameters of LLM (or use an adapter) in the MoE model? You claim that MoE performs better than P-Adapters because of the additional supervision information (in form of relations)  the model gets during the training. With the present experiments it is hard to tell. Maybe MoE perform better because you also tune parameters of LLM?



**Summary Of The Paper:**

The focus of the paper:

The paper proposes a way to increase the consistency of extracted information from the Large Language Models (LLMs) with respect to the different wordings of the queries that users use to prompt the models.

Research Directions:

There are two orthogonal directions that the authors explore. The first direction, is the use of adapters to tackle the aforementioned problem. The second direction is the reduction of supervision data that is needed to train and to test (query) the models. More specifically the authors advocate for the use of natural language prompts only. That is they do not require additional relation(s) that one can incorporate into the prompts to further specify the type of information that one can expect to be retrieved from the model.

Motivation:

The first direction is motivated by two factors: 1) efficiency - the authors aim for a lightweight approach to increase the consistency and 2) to maintain all of factual knowledge learned during pretraining.  The second direction is motivated by user experience - the authors claim that the natural language prompts are more user friendly.

Contributions:

Propose an adapter model - “P-Adapters” that rely only on natural language prompts during the training and testing. It has also been shown that the P-Adapters allows to increase consistency of the LLM to queries that are used to prompt the models.


**Summary Of The Review:**

I cannot recommend the acceptance of the paper in its present state. There are two main reasons for this. The first reason is the lack of explicitly stated  (unified) research question. Without it, it is hard to tell if the authors have conducted enough experiments to support their research question. I did my best to identify these research questions in the paper and then try to link them to the experiment they conducted. However, I had a hard time doing this. Sometimes motivation given by the authors can be ambiguous. For example, do you use adapters just for efficiency or you are interested in exploring how one can address the inconsistency problem with the adaptors?   The second reason is the lack of experiments with the frozen LLM parameters of MoE. Depending on the outcome of the experiment the conclusions that are made in the paper may change. For example: “While MoE models usually perform better, the small increase comes at the cost of requiring relation annotations at training time and specifying the subject of the entity at inference time.”

However, if those two issues are addressed, in my opinion, the paper has a good chance to be accepted. Reduction of  supervision data that is needed to tackle the inconsistency problem has both academic interest and also can be of the great interest to practitioners. Moreover, adapters are presently has high research interest in the NLP community - demonstrating how they can be used to address the inconsistency problem will also be valuable to the community.

---

> ### Author Response · Authors · 2021-11-21
> **Response to Reviewer anXM**
>
> We thank reviewer anXM for their thorough review and note that they believe our work “to tackle the inconsistency problem has both academic interest and also can be of the great interest to practitioners.” We can address Reviewer anXM’s two main concerns here.
>
> > First, Reviewer anXM identifies a number of research questions throughout the work and asks that we “please state and motivate clearly [our] (full) research question somewhere in Introduction.”
>
> The five questions Reviewer anXM identifies are:
> 1. inconsistency of LMs (in abstract)
> 2. efficiency (using lightweight prompts) (in abstract)
> 3. continuous prompts (in introduction)
> 4. user-friendly prompts (in introduction)
> 5. not updating LLM parameters maintains factual knowledge (in discussion)
>
> Our research question is 1, while 2, 3, 4, and 5 describe and motivate the setting in which we’re interested in answering this question.
>
> The reasoning is as follows (with the questions Reviewer anXM identifies above annotated in parentheses): We are interested in creating a useful system that allows users to extract factual information from LLMs. By a useful system, we mean one that works similarly to current IR systems, allowing users to express arbitrary information needs in natural language (4), and one that does not require expensive annotations. In line with previous work looking at extracting factual information from LLMs, we decide to freeze the LLM parameters and use continuous, lightweight adapters to avoid forgetting information that was learned from pretraining. We freeze the LLM parameters because, for instance, [1] finds that updating the LLM parameters can lead to lower precisions at factual probing (5). And we use lightweight adapters to interface with the frozen models and continuous prompt because both have been shown to be simple and effective in prior work (e.g., [2]) (2, 3). This all defines the setting we are interested in. The research question is then, under these constraints, how can we extract factual information accurately and consistently (1). We have updated the language in the introduction to clarify this chain of reasoning and motivation, and hope this clarifies the research question for Reviewer anXM as well.
>
> The second concern Reviewer anXM has regards additional experiments that we could run.
> > Reviewer anXM asks if we tried “to freeze the parameters of the LLM (or use an adapter in the MoE model.”
>
> We in fact do run this experiment. The LLM parameters are frozen in all settings, allowing us to make a fair comparison to the P-Adapter methods. We have clarified this in Section 4.2.
>
> > Reviewer anXM also suggests two minor experiments. Both of these involve finetuning the LLM on the factual probing task.
>
> We do not do this because other work shows that doing this can lead to forgetting information learned from pretraining [1] and suggests that one benefit of prompting a frozen model is that it maintains the linguistic knowledge of the pretrained models [2]. We have clarified this point in the introduction of the paper. That said, rigorously testing this hypothesis could be a fruitful avenue for future work, though it is beyond the current scope.
>
> If we misinterpreted either of these suggestions, we would appreciate clarification from the reviewer, and we hope in light of our responses and changes to the manuscript, the reviewer updates their recommendation.
>
> [1] Elazar, Yanai, Nora Kassner, Shauli Ravfogel, Abhilasha Ravichander, Eduard Hovy, Hinrich Schütze, and Yoav Goldberg. "Measuring and improving consistency in pretrained language models." arXiv preprint arXiv:2102.01017. 2021.
>
> [2] Lester, B., Al-Rfou, R., & Constant, N. (2021). The Power of Scale for Parameter-Efficient Prompt Tuning. EMNLP.

---

> > ### Comment · Reviewer_anXM · 2021-11-29
> > **clarifications**
> >
> > Dear Authors,
> >
> > Thank you very much for the clarifications. I am happy to increase the score.

---

### Official Review · Reviewer_v5xA · 2021-11-02

**Correctness:** 4
**Technical Novelty And Significance:** 3
**Empirical Novelty And Significance:** 3
**Recommendation:** 8
**Confidence:** 3

**Main Review:**

Pros:
+ This paper tackles one of the most important issues in the large language model: inconsistency results obtained from different prompts which have the same information needs. The problem itself is real and must be resolved.

+ The proposed adaptor, P-adaptor is a simple but effective solution to alleviate the inconsistency.

+ This paper is well-written and easy to follow. Furthermore, this paper provides comprehensive experiments including several qualitative analyses and discussions to show the effectiveness proposed method.

Concerns:
- Although the proposed method involves experiments in four different settings including OOD keyword error, it might be valuable to investigate an OOD syntax error setting. That is, in a real scenario, users write the natural language prompt which may have a grammatical error. Assessing the robustness of the proposed method in terms of grammatical errors can enhance the quality of the paper.

- In Table 2, the performances of P-tuning in RoBERTa-large are better than the ones of Oracle. Without any explanation, it is not convincing. Instead of listing the result in the tables, it would be clearer if the authors provide more explanations about them although they are in the appendix.

**Summary Of The Paper:**

This paper provides an interesting direction to extract factual information from large language models accurately and consistently. Since the quality of the factual information depends on the prompts used to query them, this paper tries to minimize the inconsistency by introducing an intermediate representation. To do this, this paper proposes an adaptor that uses the token embeddings of the natural language prompt as input and outputs continuous prompts that are used to query the large language model to extract factual information accurately and consistently.

**Summary Of The Review:**

The proposed method is reasonable and analyses of experiments are well described.

---

> ### Author Response · Authors · 2021-11-21
> **Response to Reviewer v5xA**
>
> We thank Reviewer v5xA for their review and note that they believe our work “tackles one of the most important issues in large language models” and that the solution we put forward is “simple but effective”.
>
> We appreciate the suggestion to add OOD syntax errors to the OOD settings we consider, and agree that this would be an interesting setting. We believe that the settings we currently consider are sufficient to showcase P-Adapters, so we leave this to future work.
>
> Additionally, while it might appear like the P-Tuning P-Adapter performs better than the Oracle, claiming this would be misleading as the Oracle’s performance has higher variance. (The P-Tuning P-Adapter does not achieve statistically significantly better results than the Oracle). This is due to some runs not finding optimized templates that are as effective. We have added this clarification to Figure 4’s caption in the newest draft of the paper.

---

### Official Review · Reviewer_jddf · 2021-11-02

**Correctness:** 3
**Technical Novelty And Significance:** 2
**Empirical Novelty And Significance:** 2
**Recommendation:** 5
**Confidence:** 4

**Main Review:**

Strengths:
* The problem setting--mapping natural language prompts to more optimal continuous prompts--is interesting. Most prior work on prompting attempts to find optimal prompt templates for each task (or in this setting, KB relation type), with the assumption that the task label is known at test time. Methods like P-Adapters for learning a transformation of natural language prompts could be an interesting alternative to standard prompting (with a fixed template per task) and fine-tuning, with possible applications to meta-learning/transfer learning.
* The authors compare methods with different assumptions about the availability of annotations, and under different kinds of distribution shifts.
* The paper is clearly written and the proposed methods are sensible and easy to understand.

Major comments:
* The paper does not draw a connection to prior work on robust optimization (for example, see references in [1]), which offers a more principled framework for formulating the objective of invariance to a class of input perturbations. At the very least, it would be good to give a formal definition of robustness/consistency and cite this line of prior work.
* The paper does not provide a clear justification for why P-Adapters would be expected to improve consistency, and nothing in the training objective encourages the model to be consistent. The argument might be more convincing if you could compare it to a training objective that does explicitly promote consistency--for example, by encouraging the hidden representations of different prompt paraphrases to be similar.
* As this paper notes, the factual probing benchmark has class imbalances and represents a very particular use case, so while P-Adapters appear to improve consistency here it’s unclear if these results will generalize to other settings. Would it be possible to evaluate P-Adapters on a wider variety of tasks, following prior work on prompt optimization [2, 3]? For example, [4] provide 100 prompt templates for a wider variety of NLP tasks. The paper would be more compelling if you could show that P-Adapters improve consistency in other settings too.
* Much of the motivation of this paper is based on assertions about user preferences, but there is no human evaluation to validate these claims. For example, would a typical user prefer to give a natural language input but get an inconsistent response? Or would they rather pick a pre-defined relation type with better promise of consistency? In particular, it’s not clear why user preferences would place any restrictions on training.
* MoE model: How accurate is the relation classifier? There is a big performance drop in the “OOD Prompts” setting, which leads me to wonder if the relation classifier was adequately trained. Either way, relation classification results should be included and discussed in the main paper--a perfect relation classifier would lead to 100% consistency.
* All of the models suffer a performance drop on the “OOD Objects” setting, which indicates that the models have over-fit to the (imbalanced) distribution of entities in Wikidata. It’s unclear how to interpret consistency in this setting, because the models are consistently producing the incorrect response. It might be more informative to report something like the “Consistent-Acc.” measurement from [5].
* Table 2 does not compare results to any prior work. For example, can these results be compared to the “Consistency Improved PLM” results from [5]?

Minor comments:
* The metric called “precision@1” should be called “accuracy@1”
* Table 2 does not explain how the results are aggregated (prior work takes either the micro- or macro-average over relation types).
* The description of the methods and the illustration (Figure 2) are somewhat unclear. Is there one MLP applied at each position? Three MLPs? The meaning of the different-colored tokens could also be explained in the caption.
* The paper is missing some implementation details, such as the size of the BiLSTM.
* The results section (section 5) might be easier to read if it were divided into subsections or paragraphs.
* It would be interesting to see the accuracy breakdown by relation type.

[1] Sinha, Aman, Hongseok Namkoong, and John Duchi. "Certifying Some Distributional Robustness with Principled Adversarial Training." International Conference on Learning Representations. 2018.

[2] Shin, Taylor, Yasaman Razeghi, Robert L. Logan IV, Eric Wallace, and Sameer Singh. "Eliciting Knowledge from Language Models Using Automatically Generated Prompts." EMNLP. 2020.

[3] Xiao Liu, Yanan Zheng, Zhengxiao Du, Ming Ding, Yujie Qian, Zhilin Yang, and Jie Tang. GPT
understands, too. arXiv preprint arXiv:2103.10385, 2021b.

[4] Gao, Tianyu, Adam Fisch, and Danqi Chen. "Making pre-trained language models better few-shot learners." ACL. 2021.

[5] Elazar, Yanai, Nora Kassner, Shauli Ravfogel, Abhilasha Ravichander, Eduard Hovy, Hinrich Schütze, and Yoav Goldberg. "Measuring and improving consistency in pretrained language models." arXiv preprint arXiv:2102.01017. 2021.

**Summary Of The Paper:**

This paper explores methods for improving the consistency of prompt-based factual probing of pre-trained language models. The main contributions are (1) several methods for mapping natural language prompts to continuous prompts that empirically improve accuracy and consistency on a factual probing benchmark and (2) analysis comparing reasonable alternatives in terms of accuracy and consistency.

**Summary Of The Review:**

The paper proposes methods for mapping natural language prompts to continuous prompts, with the goal of improving accuracy and consistency on a factual probing benchmark. The problem is interesting and the method appears to work on this benchmark, but I there are three main changes I would like to see before recommending this paper for acceptance, possibly at a future conference:
1. Drawing a connection to prior work on robust optimization, and providing a clearer formal justification for why this method will improve consistency.
2. Providing more detailed empirical results and discussion (in particular relation classification accuracy). The current results are hard to interpret because models can be consistent but inaccurate, and because models can perform well by over-fitting the entity distribution.
3. Ideally applying the method to a wider range of prompting tasks, to show whether the results will extend beyond the particular setting of factual probing.

---

> ### Author Response · Authors · 2021-11-21
> **Response to Reviewer jddf**
>
> We thank Reviewer jddf for their thorough review. We note they find our proposal an “interesting alternative to standard prompting.”
> Reviewer jddf is concerned with various aspects of our work. We address these here:
>
> > Reviewer jddf wanted us “draw a connection to prior work on robust optimization”, and “give a formal definition of robustness/consistency.”
>
> We agree that we have not drawn explicit connections to the vast literature on robustness to input perturbations, we have added more decision of this literature to Section 2, particularly highlighting its relevance to NLP. We also added a formal definition of consistency in Section 3.
>
> > Reviewer jddf also questions our method’s motivation, claiming “nothing in the training objective encourages the model to be consistent”, and saying that it would be “more convincing if [we] could compare to a training objective that...explicitly promote[s] consistency.”
>
> We respectfully disagree with the first point—training P-Adapters to map from variable natural language prompts to the correct object implicitly encourages consistency. Empirically we observe this with P-Adapters’s consistency outperforming the baseline. That said, we performed an additional experiment to test if explicitly promoting consistency helps and found it led to no difference in accuracy and only a small increase in consistency, suggesting the new objective was not necessary (See Appendix E).
>
> > Reviewer jddf is concerned about motivating our approach with user preferences.
>
> We respond in the overall response.
>
> > Reviewer jddf also raises questions about our results, first asking “how accurate is the relation classifier”, particularly in the “OOD Prompts setting”, and asking “if the relation classifier was adequately trained.”
>
> The classifier’s accuracy can be seen in the difference between the oracle and MoE models. We also updated the Section 5 and Appendix F to include the classifier F1 scores in each setting (and will move the table to Section 5 upon acceptance if the page limit increases). The classifier’s F1 score is 99% in ID but 81% in OOD Prompts. The high ID performance points to the classifier being adequately trained with the distribution shift explaining MOE’s performance drop.
>
> > Reviewer jddf notes that models overfit to the training object distribution, making interpreting consistency unclear in the “OOD Object” setting “because the models are consistently producing the incorrect response”, and suggests reporting “Consistent-Acc” from [5].
>
> We agree that while models overfit on the training object distribution "*our methods still outperform the baseline so what they learn is still useful for the OOD natural language prompts*." Also, we disagree that interpreting consistency is unclear in this setting: high consistency suggests that a model cannot recall a fact. We do not report Consistent-Acc because it gave similar insights to the accuracy plots. E.g., BERT Base in OOD objects:
> Baseline - 0.0; P-Adapters - (Rewrite: 0.03, P-Tuning: 0.9, Prefix: 0.06); MoE - 0.10; and Oracle - 0.24. Other models and datasets are similar.
>
> > Reviewer jddf believes “Table 2 does not compare results to any prior work”, and asks if we can compare to [5].
>
> We write: *The oracle…serves as a...comparison to P-Tuning [prior work] because P-Tuning assumes access to the gold relation* at inference time [1] (Section 4.2). Unfortunately, we cannot compare to [5] because their vocabulary is restricted to a relation’s valid entities, while ours is the intersection of RoBERTa and BERT vocabularies.
>
> > Reviewer jddf also wonders if it would “be possible to evaluate P-Adapters on a wider variety of tasks” because “the factual probing benchmark has class imbalances and represents a very particular use case.”
>
> We agree that factual probing is a particular use case: one with high entropy as models can be queried for many different fact types with user prompts. Prompting for other NLU tasks (e.g. sentiment) doesn’t require interfacing with user prompts: one can just use a predefined prompt that specifies the task well. This is what makes factual probing interesting, and therefore, we don’t expect P-Adapters are required for good performance on many other tasks prior work has considered. We defer such investigations to future work.
>
> Minor comments:
> - We use precision to match the LAMA probe paper [2].
> - We report results microaveraged over relation types and independently apply a single MLP to each hidden state.
> - BiLSTM size (384/direction) is now in Section 4.2
> - The breakdown of results by relation type is not too interesting. Most relations show the same pattern as the aggregate, though five are much more difficult than others with all methods fail to get precision above 15%.
>
> We would appreciate if Reviewer jddf reconsider their score in light of these new experiments and clarifications.
>
> [1] Liu, X., et al., (2021). GPT Understands, Too.
>
> [2] Petroni, F., et al., (2019). Language Models as Knowledge Bases?

---

> > ### Comment · Reviewer_jddf · 2021-11-29
> > **Second response**
> >
> > Thank you for the clarifications, the updates in sections 2 and 3, and the additional experiments. I'm willing to update my score to 5: "marginally below the acceptance threshold". The reason I'm still leaning to reject the current version of the paper is because I still have two main concerns:
> >
> > In response to the question, would users "rather pick a pre-defined relation type with better promise of consistency?", the authors respond:
> > > This trade-off is not one that we consider because we do not consider the 41 relations to span all possible (or even a large number of) information needs. (In fact, one of the benefits of P-Adapters is that they do not depend on a user’s query being one of the 41 it was trained with like the MoE or oracle models do.)
> > This claim would be more convincing if it was supported by experiments testing the model on out-of-domain relations. From what I understand, all of the OOD settings considered in this paper use the same 41 relations for training and evaluation, but with different distributions of prompts or objects. From these experiments, it's hard to conclude that these methods would improve consistency on unseen relations.
> >
> > In response to my questions about the accuracy of the relation classifier, the authors respond:
> > > The classifier’s F1 score is 99% in ID but 81% in OOD Prompts. The high ID performance points to the classifier being adequately trained with the distribution shift explaining MOE’s performance drop.
> > The additional discussion in Section 5 / Appendix F is helpful, although I'm still not convinced that this is the strongest version of the MoE baseline, because the classifier overfits to the specific in-domain templates. From my understanding, ID score refers to F1 on prompts with the same templates but different subjects or objects--but you could get 100% ID accuracy on this classification task just by doing substring match on the training templates. I think a better training setup would involve holding out some prompt templates while training the classifier, to monitor against over-fitting.

---

> > > ### Author Response · Authors · 2021-12-03
> > > **Response to jddf**
> > >
> > > We appreciate Reviewer jddf’s willingness to update their score and ask if the reviewer can change the score in the original review to reflect this.
> > >
> > > In response to the concerns the reviewer raises. First, we agree that evaluating on unseen relations would be a good way to further test our system, and is a good next step for future work. Second, the reviewer believes that the MoE baseline could be made stronger. We appreciate the suggestion to hold out some templates as a validation set—we had monitored for overfitting using a held-out set of subject-object pairs, so this would be a good extension. We would still expect to see some drop because the OOD Prompts are different than the ones seen during training, but such a method might decrease the difference.
> > >
> > > That said, we want to push back a bit on the idea that the relation classifier does poorly in-domain. We don’t think that we can “get 100% ID accuracy on the classification task just by doing substring match on the training templates”. There are some relations that have identical templates, and disambiguating the relation requires some additional world or syntactic knowledge. For example, the ID templates include both “Ottawa is the capital of [MASK].” and “[MASK] is the capital of Canada.” These are considered two separate relations in the dataset, so the classifier must distinguish between them. Examples like these are likely why we don’t see 100% ID accuracy.
> > >
> > > Again, we wanted to thank the Reviewer jddf for engaging so thoughtfully with our work, and ask if they can update the score in their original review.

---

> > > > ### Comment · Reviewer_jddf · 2021-12-03
> > > > **Updated score**
> > > >
> > > > I updated the score in my original review.

---

### Official Review · Reviewer_2hiF · 2021-11-05

**Correctness:** 3
**Technical Novelty And Significance:** 3
**Empirical Novelty And Significance:** 3
**Recommendation:** 6
**Confidence:** 3

**Main Review:**


Authors also presented a Mixture of Experts models that learn a set of prompts and select one to query the LLM. Experimenta results show that P-Adapters perform comparably to the more complex MoE models in extracting factual information from BERT and RoBERTa while eliminating the need for additional annotations.

The model and training procedure are well described and results are promising. It is also great that the data used is in the Github repositories. This said, the reader could benefit from better error analysis, as it is not clear how much hallucination and grounding the model is producing.


**Summary Of The Paper:**

The paper presents an approach for extracting factual information from LLMs, where authors discuss a user oriented setting where they take in varied natural language prompts and return the objects with information needs. Authors proposed an approach, denoted P-Adapters, which is a model that is between the embedding layer and first attention layer of LLMs. It takes LLM embeddings as input and output prompts used to query the LLM.

Authors also presented a Mixture of Experts models that learn a set of prompts and select one to query the LLM. Experimenta results show that P-Adapters perform comparably to the more complex MoE models in extracting factual information from BERT and RoBERTa while eliminating the need for additional annotations.

The model and training procedure are well described and results are promising. It is also great that the data used is in the Github repositories. This said, the reader could benefit from better error analysis, as it is not clear how much hallucination and grounding the model is producing.


**Summary Of The Review:**

The obtained results and model itself will benefit the reader and researchers working in this important research area.

---

> ### Author Response · Authors · 2021-11-21
> **Response to Reviewer 2hiF**
>
> We thank Reviewer 2hiF for taking the time to review our paper and are glad they find that our “results are promising”. Reviewer 2hiF also believed that our “model and training procedure were well described.”
>
> > Reviewer 2hiF believed that our work could benefit from better error analysis” to see “how much hallucination and grounding the model is producing.”
>
> We conduct some additional qualitative error analysis and add it to Appendix Section D. As a summary, the difference between the P-Adapters and the Baseline and MoE models comes down to the consistency of the prompts. All make some mistakes, while the P-Adapters make more consistent and accurate predictions than the baseline, they are less consistent than the MoE model. This can actually be beneficial in cases where the MoE model predicts the same, incorrect object for multiple paraphrases but the P-Adapter splits its predictions among the incorrect and correct object. We also see some evidence of P-Adapters being more sensitive to surface form flexibility than the MoE models (for example, incorrectly predicting Akihiko Saito died in Tokyo, rather than Iraq). We encourage Reviewer 2hiF to see further details in the new Appendix Section D.

---

### Official Review · Reviewer_AfJQ · 2021-11-05

**Correctness:** 4
**Technical Novelty And Significance:** 3
**Empirical Novelty And Significance:** 3
**Recommendation:** 8
**Confidence:** 4

**Main Review:**

### Strengths of the paper:

- The problem of extracting factual and consistent information from large language models is of high interest to the NLP community. Given how LLMs dominate NLP at the moment, making sure these models are robust and consistent is a timely problem,
The paper is overall well written, with only a couple of confusing parts (see below),

- The proposed architecture for intervening between the input embeddings and the first hidden layer of the language model is quite comprehensive. I enjoyed seeing the different options, and in particular, thought the use of the MoE for relation classification to be quite insightful,

- The experimental analysis of the work is well executed, and demonstrated convincingly which interventions were most useful in make predictions more accurate and consistent,

- I liked the analysis in Figure 6, showing the importance of the subject entity on the precision of the fact extraction task,

### Weaknesses of the paper:

- The main weakness in this work is one that relates to the overall goal of fact extraction from language models. The “Oracle'' results from Table 1 are thought provoking: with perfect knowledge regarding the predicate/relation of test examples, and a subsequent 100% consistent response, the LLM is only able to obtain ~50% correct responses from T-Rex, which is an admittedly limited evaluation (41 “head” predicates, mostly of well known entities). While I understand that this work is clearly focused on the consistency issue, not necessarily correctness, it puts into question whether fact extraction from LMs is a worthwhile pursuit.

- I would have liked for the paper to dig a little deeper into this headroom question from the previous point.  Would it be possible to conduct a sampled qualitative evaluation of errors of the Oracle model in the ID cases?  Are the errors due to unseen triples during training time (e.g., not in Wikipedia), or maybe there are issues with model capacity (maybe a 10x version of the LM would be able to recall the prompted fact)?

- In terms of writing, the most confusing section in the paper is Section 4.1. After re-reading it twice, I was still not able to ascertain: (1) what data was used to train the models, and (2) what data was used to evaluate the models. The section makes reference to LAMA’s T-REX, LPAQA, ParaRel, as well as augmentations using BERT lexical replacements, as well as data from “Shin et al, 2020”. The section also talks about examples from these sources as well as templates (presumably filled in with WikiData triples?). I really think this section needs to be rewritten and the training, eval and test datasets should be much more precisely described.  I would also encourage authors to release the exact datasets and splits to allow others to reproduce/improve on this work. But even with a data release, a precise description of how this data was constructed is very important.

- For the MoE and Oracle layers, the description in the paper is insufficient to determine the outputs presented to the first layer of the model. The depiction in Figure 2 hints that the entire sequence is rewritten using the fixed-length learned embeddings, and perhaps the subject or MASK embeddings are preserved? But actually sub-section 4.2 never formally describes how the embeddings are used to create the continuous prompts? Are they prepended/appended to the original inputs? Or do they rewrite the original inputs? Do either the MASK or subject tokens get copied?

- The LAMA benchmarks have one unfortunate characteristic: since it was constructed for BERT-style single token prediction, it has stripped down the original datasets (see the original version of T-Rex, which contains over 600 unique predicates vs. the 41 from LAMA: https://hadyelsahar.github.io/t-rex/  and https://aclanthology.org/L18-1544.pdf ). I wonder if a more comprehensive version of this would be to evaluate on a larger sequence-to-sequence model like BART https://arxiv.org/abs/1910.13461  or T5 https://arxiv.org/abs/1910.10683 (both available as HuggingFace models). Given that this work leverages frozen LLMs, it seems that training and evaluation could be done relatively cheaply even for larger models with proper decoders.


### Other comments:

- With respect to the MoE solution, the paper claims that the model does not use a weighted combination and opts to use the top-1 predicted relation. I wonder if authors have tried using a weighted combination instead? If the relation classifier is trained with cross-entropy softmax loss, most of the weights will be close to one-hot (similar to top-1) except when the model is uncertain. Therefore combining prompt embeddings may yield some benefit over top-1. Does this make sense?

- Note sure this is a good idea, but: given that the LLM is frozen, it seems plausible that the continuous prompt embeddings learned in some of the models resemble existing embeddings from the original vocabulary. As such, would it make sense to attempt to “decode” the continuous prompt embeddings into the existing vocabulary? One could use a greedy decoding strategy of extracting the nearest neighbor (via dot product or cosine distance) from each continuous prompt embedding to the vocabulary input embedding table. Have the authors tried  inspecting the continuous prompts in this way? I wonder if the output is informative or whether these prompts are modeling purely latent variables.

- Typo in Figure 1 “Canada si” -> “Canada is”,

- Typo in page 6: “Cannonical”  -> “canonical”


**Summary Of The Paper:**

This paper addresses the problem of robustness for extracting factual information from large language models. It first describes and motivates the problem of inconsistent predictions of large language models on fact-seeking prompts when these prompts are perturbed or rephrased.  It then proposes a few different methods for addressing this inconsistency that operate on the same portion of the language-model, namely, between the input token embeddings and the first hidden layer of the language model. The work evaluates the performance of the variants using a pooled collection of fact-seeking prompts (e.g., LAMA, LPAQA and ParaSel). The results employ a consistency metric and show that different interventions in the input embeddings cause large differences in inter-prompt consistency.


**Summary Of The Review:**

The problem of extracting factual and consistent information from large language models is of high interest to the NLP community, and this work in particular should be of interest to the ICLR community. Overall, this work was well-written throughout (easy to follow in most places except for a few rough parts detailed above). The experimentation work was also of high quality, with interesting results. To highlight a few findings: (1) the use of a relation-classification MoE and its consistently high performance on consistency metric seems promising, (2) the analysis demonstrating the importance of the “subject” is correct fact prediction, and (3) analysis demonstrating the negatives effects of uniformizing objects in train/test sets, which is strong indication that LLMs still do not generalize well to unseen objects.

---

> ### Author Response · Authors · 2021-11-21
> **Response to Reviewer AfJQ**
>
> We thank Reviewer AfJQ for their detailed review and for describing our work as of “of high interest to the NLP community”. Reviewer AfJQ  found our proposed architecture to be “quite comprehensive”, and “thought the use of the MoE for relation classification to be quite insightful.” Reviewer AfJQ also believes the “experimental analysis of the work is well executed, and demonstrated convincingly which interventions were most userful”.
>
> Reviewer AfJQ also offers some concerns, which we respond to below:
>
> > Reviewer AfJQ is concerned that with the oracle methods, “the LLM is only able to obtain ~50% correct responses” and wonders if “the errors are due to unseen triples during [pre]training time” or due to “issues with model capacity”.
>
> We hypothesize that it is a model capacity issue. The models saw all of the LAMA entity triples we ask them to identify during pretraining. The triples are taken from Wikipedia and are annotated with subsequences of wikipedia passages where they are expressed, and the pretraining corpus for BERT and RoBERTa does include WIkipedia. Reviewer AfJQ also suggests that we could “conduct a sampled qualitative evaluation of errors in the Oracle model in the ID cases”, but also acknowledges that “this work is clearly focused on the consistency issue, not necessarily correctness.” We agree that this sort of investigation would be quite interesting, but is outside the scope of the current work.
>
> > Reviewer AfJQ also found Section 4.1 to be “the most confusing section in the paper”, and is concerned that “sub-section 4.2 never formally describes how the embeddings are used to create the continuous prompts”
>
> Following Reviewer AfJQ’s recommendation, we have re-written Section 4.1 to make precise the data setting we consider and added additional connections between the the P-Tuning P-Adapter and MoE/Oracle settings that were missing from Section 4.2. We also point to Section 3, explaining the terminology of “entity pairs” and “templates”. To directly answer the questions posed regarding the MoE and Oracle methods:
> > “perhaps the subject or MASK embeddings are preserved?” Do either the MASK or subject tokens get copied?”
>
> Yes, the subject and MASK embeddings are preserved and are copied from input to output.
>
> >“how the embeddings are used to create the continuous prompts? Are they prepended/appended to the original inputs? Or do they rewrite the original inputs?”
>
> The original inputs are ignored and the continuous prompts depend only on the relation.
> We will release the data on github pending acceptance, and Reviewer AfJQ can find it in the supplementary material as well.
>
> > Reviewer AfJQ also proposes intriguing avenues for future work: wondering if we could “evaluate on a larger sequence-to-sequence model like BART or T5”; using a “weighted combination” of the relation classifier; and wonders if it “would ...make sense to attempt to “decode” the continuous prompt embeddings into the existing vocabulary?”
>
> Our method could very well be extended to testing on sequence-to-sequence models. There is somewhat less precedent for this in factual extraction which is why we do not perform this. It is definitely a potentially fruitful avenue for future work.
>
> As for the weighted combination, this makes sense, but as the reviewer observes these probabilities would mostly be close to one-hot vectors and would likely not be calibrated, so we doubt we would see much benefit. We can perform these evaluations if they are strongly recommend and report the results in a later version.
>
> Looking at the similarity between the learned prompts (the P-Adapter outputs) and the embeddings is an interesting idea. Other work that has looked at similarities between embeddings have found that the most similar embeddings to learned prompts are not that interpretable [1].
>
> [1] Qin, G., & Eisner, J. (2021). Learning How to Ask: Querying LMs with Mixtures of Soft Prompts. ArXiv, abs/2104.06599.

---

### Author Response · Authors · 2021-11-21
**Response to All Reviewers**

We thank the reviewers for all their thoughtful comments. The consensus is that this work contributes to an important area of research, and many reviewers had insightful suggestions for future work. Reviewers AfJQ, 2hiF, and v5xA all were in favor of accepting the paper and Reviewer anXM made their acceptance conditional on two points which we have clarified. Reviewer jddf was in favor of rejecting, and we have provided substantial evidence to address their concerns in our response.

We address one concern that Reviewers anXM and jddf both had here and address other concerns in the responses to the individual reviewers.

> Reviewers anXM and jddf both had concerns about our user-centric motivation. Specifically, Reviewer jddf had concerns with claiming we were motivated by user preferences without performing “human evaluation to validate these claims”. Reviewer jddf poses a trade-off asking “would a typical user prefer to give a natural language input but get an inconsistent response? Or would they rather pick a pre-defined relation type with better promise of consistency?” Reviewer anXM asks if we find work that “compares the user experience with and without the additional information”

This trade-off is not one that we consider because we do not consider the 41 relations to span all possible (or even a large number of) information needs. (In fact, one of the benefits of P-Adapters is that they do not depend on a user’s query being one of the 41 it was trained with like the MoE or oracle models do.)
Our starting assumption is that users have arbitrary information needs, and that these are best expressed through natural language queries. This assumption originates from the design of modern search engines, which accept natural language queries, and the expressivity of language. Ours and prior work has noted that expressing information needs through natural language queries leads to issues with consistency when querying LLMs, and our work is meant to help mitigate these inconsistencies. What situations users would value consistency so much as to prefer choosing the relation is potentially an interesting question, but is outside the scope of the current work.

We thank all the reviewers for their time and suggestions!

---

### Decision · Program_Chairs · 2022-01-20

**Decision:**

Accept (Poster)

**Comment:**

This paper introduces a prompting technique for eliciting factual knowledge from frozen pretained transformer LMs. The key idea is to modify the embeddings produced by the embedding layer before they are passed to the first attention layer and the paper investigates several different design choices. The Reviewers all agree that the paper tackles an important problem with interesting methods, that it is well written and has strong results. The main concerns, raised by Reviewer jddf, were about clarifying the connections to the robust optimization literature and evaluating on OOD relations. The former has been addressed in the revised version. While the latter point remains valid, I find that the paper in its current state has enough useful experiments and analysis to warrant publication. The authors have clarified most of the other points raised by the reviewers in their rebuttal.